# The Pour-Through Procedure for Monitoring Container Substrate Chemical Properties: A Review

**James E. Altland**

United States Department of Agriculture, Application Technology Research Unit, Wooster, OH 44691, USA; james.altland@usda.gov

**Abstract:** The pour-through procedure is a nondestructive method commonly used by horticultural crop producers and research scientists to measure chemical properties and nutrient availability in container substrates. It is a method that uses water as a displacement solution to push the substrate solution out of the bottom of the container so it can be analyzed for pH, electrical conductivity, and nutrient concentrations. The method was first introduced in the early 1980s. Since then, research has been conducted to determine factors that affect the results of the pour-through including volume, nature and timing of application of the displacement solution, container size, and substrate stratification. It has also been validated against other common methods for determining container substrate pH, EC, and nutrient concentration, most notably the saturated media extraction procedure. Over the past 40 years, the method has been proven to be simple, robust, and consistent in providing crop producers and researchers valuable information on substrate chemical properties from which management decisions and experimental inferences can be made.

**Keywords:** soilless substrates; nursery; floriculture; fertilization

## 1. Basic Premise

The pour-through procedure, also referred to as the Virginia Tech Extraction Method [1,2] or the PourThru method [3], is one of the most utilized techniques for assessing the nutritional status of container-grown plants. Deionized water is used to displace an aliquot of the substrate solution used as a proxy of the nutritional status in the container [4]. Testing of field soils usually involves the measurement of plant nutrients sorbed to soil colloids as well as those in the soil solution. However, the relatively high mineral nutrient intensity and low ion exchange capacity of soilless substrates dictates that the bulk solution is more critical as an instantaneous gauge of the container nutritional status [5]. By using the pour-through procedure to assess the chemical properties of the pore water, growers can make decisions to quickly adjust crop fertility, and researchers can assess the availability of mineral nutrients or make inferences on the impact of experimental treatments on the substrate.

The procedure involves saturating container(s) to their water-holding capacity via normal irrigation, then waiting 30 to 60 min for the substrate solution to come to equilibrium both in its physical movement through the container pore spaces, and in its chemical properties as the solution interacts with fertilizers and the substrate itself. After this equilibrium is reached, a small volume of water is poured over the container surface (displacement solution) thus forcing a portion of the substrate solution to move out of the drainage holes at the container bottom. The leached substrate solution is captured, measured immediately for pH or EC, or stored for later analysis of mineral nutrients [6].

## 2. History of the Pour-Through Procedure

The pour-through procedure is often credited to Yeager et al. [7], although they cite the procedure being used in previous publications by their group [8,9]. Some extension articles and trade publications even refer to the procedure as the Virginia Tech Extraction Method

(VTEM) [1,2,10], crediting the university in which the three aforementioned publications were written. However, this was not the first time researchers used effluent to measure substrate chemical properties. Hipp et al. [11] compared three methods to the saturated paste procedure, considered the standard method of the time, for monitoring pH in small containers filled with media. The three alternate methods considered were to use an in situ ceramic cup extractor, monitor effluent, and place the pH probe directly in the media. The measurement of effluent was conducted by collecting the first 250 mL of leachate after an irrigation event. The volume of water applied was not disclosed, only that enough water was applied to generate 250 mL of leachate. This differed from the methods used by Yeager et al. [7] in that the substrate was not saturated before irrigation was applied to the media to generate the effluent. Despite this, the authors observed high correlation (R = 0.95) between the saturated paste procedure and their effluent procedure. Jarrell et al. [12] also measured pH of leachates. They irrigated 10 $cm^2$ pots enough to produce 10 mL of leachate and measured the pH of that solution. They described good agreement of leachate pH with saturated media extraction (SME) pH (without statistics) but warn that leachate pH might be affected by plant type. Similar to procedures described by Hipp et al. [11], these authors collected leachate directly from irrigation events. Schoonover and Sciaroni [13] also describe an experiment by which substrates composed of fine sand and either peat or fir bark were leached to remove salinity, and leachates were collected periodically to determine EC. The same authors [13] described a SME procedure as being the standard and preferred method for monitoring substrate pH and nutrition levels.

Collecting leachates for measurement is not unique to container-grown plants. Thirty years prior to the work by Yeager et al., [8], Krone et al. [14] described the common practice of placing a pan in soil to capture downward moving water through the soil profile. Wagner [15] also summarized the use of lysimeters, essentially large tubes filled with soil that are analogous to containers filled with substrates, to measure the quantity and quality of water moving through a soil profile. While both methods were common during this time, both authors described the problems associated with collecting drainage water from soils artificially suspended in a tube or container. Burd and Martin [16] provided perhaps the first thorough assessment on the feasibility of using a displacement solution to extract soil solution from field soils packed into columns.

While the function of soil lysimeters and work by Hipp et al. [11] and Jarrell et al. [12] probably influenced the work of Yeager et al. [7], the pour-through procedure used today is distinct from these earlier procedures and can be tied directly to the method described by Yeager et al. [7]. The primary characteristic that distinguishes the work of Yeager et al. [7] from all that had been published previously is that it dictates to first saturate the substrate and allow it to drain prior to extraction. A period of time (not defined) is then provided for the substrate solution to come to equilibrium with the salts and media within the container substrate. Only then is a small volume of displacement solution poured on the substrate surface to push a portion of the substrate solution out of the container bottom for analysis. The primary advantage of this saturation step is that it reasonably assures a similar moisture content each time the procedure is used. Cavins et al. [3] showed that mass wetness of the substrate, as well as leachate volume, decreased between 120 and 240 min after containers were saturated, likely due to evapotranspiration from the substrate. Yeager et al. [7] demonstrated that EC and nitrate concentration increased as moisture levels decreased. This might seem counterintuitive, as one would expect higher salt readings as moisture content decreases and salts become more concentrated. However, Yeager et al. [7] suggested that greater moisture content facilitates increased micropore saturation of the substrate particles, resulting in greater ion diffusion and nutrient extraction. Saturating the containers prior to each pour-through measurement is a critical step in the overall procedure to ensure consistent moisture content in the substrate, thus the contribution by Yeager et al. [7] in establishing the method as it is currently used is substantial, especially considering the common practice of measuring plants or a group of plants repeatedly over time to track crop fertility throughout production.

### 3. Factors That Affect the Pour-Through Procedure

*3.1. Volume of Displacement Solution versus Container Size*

The volume of the displacement solution is one of the most critical steps in the pour-through procedure. In a fully saturated container, any volume of displacement solution applied to the container surface should displace the same volume of substrate solution from the bottom of the container. Ideally, the displacement solution would not mix with or alter the substrate solution, but only push it like a piston through a cylinder; however, under normal conditions the displacement solution moves unevenly through the substrate (relatively unrestricted through macropores and restricted through micropores) and mixes with the substrate solution [17]. A sufficiently representative volume of substrate solution should be collected to provide confidence that the solution is representative of the entire container but that also minimizes the chance of the displacement solution contaminating the leached substrate solution. Most current recommendations suggest that a leached volume of 50 mL is sufficient [2]. Nuñez and Osborne [18] applied sufficient volume to generate either <50 mL, 50 to 150 mL, or >150 mL of leachate solution. Leachate pH and EC tended to decrease with increasing volume applied, however, they did not provide any analytical evidence that one or a range of leachate volumes provided more accurate results than the others. Yao et al. [19] applied 40 to 120 mL of displacement solution to the top of 10.5 cm pots filled with sphagnum peat substrate and found that leachate volume increased proportionally with increasing displacement solution volume, EC declined when the displacement solution volume exceeded 90 mL, but pH remained constant over the range of applied volumes. In the most comprehensive analysis of displacement solution volume relative to container volume, Torres et al. [20] compared pour-through volume on the results of pH and EC by applying enough water to extract 50 mL or 2.5% of container volume with containers that were either 2, 8, 11, or 27 L in volume. Across all container sizes, they [20] found that collecting a 50-mL sample provided more consistent results for pH and EC compared to collecting 2.5% of the container volume. LeBude and Bilderback [2] provided a table of recommended displacement solution volumes for container volumes ranging from 4 to 95 L.

*3.2. Nature of the Displacement Solution*

Many guidelines for conducting the pour-through procedure dictate the displacement solution should be deionized or purified water [7,21]. However, if the displacement solution is applied so that it pushes substrate solution from the bottom of the container without mixing or contaminating the displacement solution, then the qualities of the displacement solution should not affect the results from the pour-through procedure. Yao et al. [19] found that displacing solutions with EC ranging from 0.001 to 0.93 dS·m$^{-1}$ did not affect pH or EC of the measured pour-through leachates.

*3.3. Timing of Displacement Solution Application*

Cavins et al. [3,4] examined the time between the saturation step and application of the displacement solution. Leachates were collected via the pour-through method at 15, 30, 60, 120, and 240 min after the saturation process. They found volumetric water content and leachate volume were reduced after 120 min following saturation; however, despite a reduction in leachate volume there was no effect on pH or EC. They [3] concluded that the displacement solution should be applied 60 min following saturation to allow for equilibration of the substrate solution but avoid potential artefacts from reduced substrate moisture. Yao et al. [19] collected samples from 0 to 160 min after saturation and found that pH and EC did not change between 20 and 120 min. Compton and Nelson [22] examined the time after fertilization or irrigation for collecting substrate solution from plug seedling trays by squeezing the rootball through cheese cloth (known as the squeeze method). While this method for extracting the substrate solution differs from the pour-through method, they also concluded that 60 to 120 min is the ideal time after a fertilizer application to collect a sample.

### 3.4. Substrate Type

Niemiera et al. [23] demonstrated that substrate type can affect interpretation of pour-through results. They used 100% pine bark, 9 pine bark:1 sand, and 5 pine bark:1 sand substrates with 84%, 75%, and 66% total porosity, respectively. They concluded that greater porosity and drainage in 100% pine bark substrate allowed greater channeling of the displacement solution through the substrate, resulting in lower nitrate concentrations in the leachate due to dilution. Yao et al. [19] argued that sphagnum moss, with air-filled porosity and cation exchange capacity (CEC) different from peatmoss or pine bark, could affect PT results and interpretation. While they [19] did not directly compare the sphagnum moss to other substrates, they concluded that the pour-through procedure with sphagnum moss using a variety of displacement solutions between 20 and 160 min after saturation should provide similar results as other substrates, and thus those standards could be applied.

### 3.5. Substrate Stratification

Virtually all containers in commercial production of floriculture and nursery crops are filled uniformly with a single mixed substrate. However, the substrate does not remain uniform after irrigation. Opposing forces from the gravitational potential gradient and capillarity create a zone of saturation called the perched water table at the container bottom [24], with decreasing volumetric water content with increasing height above the perched water table [25]. This volumetric water gradient can also affect other chemical and biological properties as greater moisture affects decomposition processes [26], mineral nutrient release from controlled release fertilizers [27], and gas emissions from the substrate [28]. Wada et al. [29] showed decreasing substrate pH in containers from the surface to the container bottom and attributed the higher pH near the substrate surface to irrigation water alkalinity having greater effect on the substrate surface than deeper in the container. Jeong et al. [30] also reported stratification of pH and EC through the container profile; however, they reported the opposite trend with pH increasing from the top to the bottom while EC decreased. They [30] attributed lower pH near the surface to fertilizer solution causing acidification reactions near the surface and less so deeper in the container profile. Others [31,32] reported results similar to Jeong et al. [30] and further speculated that accumulation and nitrification of ammonium near the surface from application of water-soluble fertilizers are responsible for reduced pH. Molitor [33] similarly showed reduced pH in the lower portion of the container when fertilizer solution was applied via subirrigation. Handreck [31] demonstrated the pour-through method provides pH and EC values closer to the pH and EC of the bottom strata of the container, thus highlighting the problem with interpreting results from the PT method. He concluded that that containers should be checked (without suggesting how) for the absence of large vertical variation in pH or EC before pour-through extracts can be considered to be a reliable substitute for the SME procedure.

## 4. Comparison with Other Procedures

Numerous methods have been proposed for monitoring the nutritional status of container substrates including the use of suction cup lysimeters [34], petiole sap tests [35], various water to substrate extraction ratios [36], the press test [19], and the perched water displacement method [37]. However, the SME and pour-through procedure remain the most commonly used and referenced procedures and are the two with the most well-established guidelines for plant production [4,19,38], with the pour-through providing the advantage that it does not require destructively harvesting the container to obtain the substrate solution.

The pour-through and SME procedures have been shown to provide similar values in substrate pH, but differing values in EC and other dissolved nutrients. Cavins et al. [4] reported no differences in pH measured by the pour-through or SME procedures on 16.5-cm pots filled with a sphagnum peat and pine bark substrate. However, they reported EC, nitrate, phosphorus, potassium, calcium, and magnesium levels 1.4 to 1.6 times higher

for the pour-through procedure compared to the SME. Wright et al. [38] reported similar pH between the two methods in 15-cm containers filled with a sphagnum peat, pine bark, and vermiculite substrate over the range of 5.6 (the lowest measured value) to 7.0. Similar to Cavins et al. [4], Wright et al. [38] reported that EC and all other nutrients measured were higher with the pour-through procedure compared to the SME, although correlation analysis between the two procedures was not used so that a conversion factor cannot be readily discerned. Yeager et al. [7] also reported nearly identical pH measurements from pour-through and SME procedures but higher EC and nutrient levels. However, similar to Wright et al. [38], they provided correlation analyses between each method and fertilizer application rates. They did not provide correlation analyses between the two methods. While plotted figures provided a relatively clear relationship between the two methods, no statistical analyses were provided to offer a mathematical relationship.

Similarities between the pour-through and the SME procedure for pH measurements while simultaneously providing different EC or nutrient levels are likely a function of dilution. Subjecting containers to the pour-through procedure will force approximately 50 mL of substrate solution from the bottom of the container without dilution. In contrast, subjecting container substrates to the SME procedure requires additional deionized water be added to the substrate (usually within a glass beaker or jar) until all pore spaces are filled, resulting in 30% to 50% more water being added to the substrate solution based on the average air space of pine bark substrates [39]. The additional deionized water added to the SME procedure explains the 30% to 40% reduction in SME values compared to pour-through values [4,40].

Alternate methods that extract an undiluted solution from the substrate have been shown to provide similar results to the pour-through procedure. For example, Yao et al. [19] found that the pour-through method was highly correlated (R = 0.97) to the press method with EC ranging from 0 to 1.6 dS·m$^{-1}$. The press method involved squeezing 10.5-cm pots until about 30 mL leached through the container bottom and thus measured the substrate solution directly (without dilution) similar to the pour-through method. Likewise, Jeong et al. [30] showed that solutions extracted with a Rhizon soil moisture sampler provided similar pH and EC values to the pour-through method.

## 5. Measurement of Specific Nutrients

The pour-through procedure is an effective way to measure substrate pH, electrical conductivity, and macronutrient concentration in the substrate solution. However, because it uses water as a displacement solution to "push" the substrate solution from the bottom of the container, it is not as effective in displacing micronutrient cations (Fe, Mn, Cu, and Zn) that are present in low concertation in the substrate. Numerous chemical extractants have been compared to water as an extractant for the SME procedure including acids, bases, boiling water, and chelates [36,41–43]. Berghage et al. [42] showed that a 5 mM solution of diethylenetriaminepentaacetic acid (DTPA) provided the greatest resolution of micronutrient concentrations compared to water and 13 other acids and chelates when extracting substrates with three different levels of micronutrient amendment. Micronutrient concentrations from water were generally too low to accurately differentiate between micronutrient levels in the substrate. Using water as the displacement solution in the pour-through procedure has a similar and inadequate ability to 'extract' micronutrient cations from the substrate for analysis in the solution.

## 6. Conclusions and Recommendations

The pour-through procedure is a simple, robust, and scalable procedure that can be utilized by practitioners and researchers to provide accurate measures of pore water pH, a good approximation of EC and macronutrient concentrations, but limited resolution for monitoring micronutrient concentrations. Based on research published thus far, the procedure can be effectively applied to containers of almost any size and filled with any substrate. The procedure should be applied to containers that have been saturated and

allowed to come to equilibrium for approximately 60 min. A displacement solution of tap water should be applied slowly, distributed evenly across the substrate surface avoiding container walls where channeling may occur, and in a manner that avoids mixing and contamination with the leached substrate solution. As with many analytical procedures, when conducted consistently over time with attention to details, the results provide meaningful data from which action (changes to a fertilizer program, for example) or inferences (in research) can be made.

Future research could evaluate repeated measurements of the same container over time, and whether or not this affects pH and EC readings after multiple successive pour-throughs. The impact of stratified layers within the container caused by the moisture gradient or intentional production practices (such as layering of controlled release fertilizers) should also be addressed. Ideally, the pour-through procedure would provide a sample that represents the entire bulk of the substrate, not only the bottom layer. Research is also needed to determine how the pour-through procedure could be modified to provide more resolution for measuring micronutrient concentrations in the substrate.

**Funding:** This research received no external funding.

**Institutional Review Board Statement:** Not applicable.

**Informed Consent Statement:** Not applicable.

**Conflicts of Interest:** The authors declare no conflict of interest.

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
