# Peer review of "The Pour-Through Procedure for Monitoring Container Substrate Chemical Properties: A Review"

_horticulturae, doi:10.3390/horticulturae7120536_

Round 1

Reviewer 1 Report

This is a nice paper and is well-written. Please consider including guidelines on interpreting pour-thru results, and how the interpretation compares to that of SME and 2:1.

I think this article would greatly benefit from additional of data and tables/figures that help summarize the general concepts outlined in the text with values and ranges\.

Author Response

I appreciate the effort of the reviewers and the comments made to improve the manuscript.  Below I have my response to the Reviewers.  However, if the Editor feels strongly that I should take a different course of action, I will make those changes promptly.

Also, in reviewing the paper once more, I found two places where I provide very minor changes to the text.  I used ‘Track Changes’ so those can be found easily by the Editor.

Reviewer 1 requests tables for comparing the results from the pour-through to the SME.  I appreciate the suggestion and think it is a good one.  However, as written in the discussion, several papers have made comparisons between the two procedures but the methodology differs for each paper.  Describing how these procedures relate to each other really requires a discussion, not a table or figure.  I believe a table or figure comparing the two procedures would lack the nuance and specificity needed.  I respectfully suggest that a discussion, as it is currently written, is a better way to compare these procedures.

Reviewer 2 Report

I have revised the manuscript entitled "The Pour-Through Procedure for Monitoring Container Sub-2 strate Chemical Properties: A Review". The work is a review about the pour-through procedure to measure chemical properties and nutrient availability in container substrates. I can not issue a recommendation about the manuscript.

The work is properly made, with all the parts correctly developed, a deep review about the topic is done. To me, the review is properly done and all previous related works are accordingly cited and refered. However, due to the controversy exposed into the doccument (process credited to Yeager et al.). I do not feel expert enought to understand if the procedure is properly credeted or not. From technical aspects, the manuscript is correct, but I suggest to ask for review to a top author in the topic, with a better criteria to solve this problem.

Author Response

I appreciate the effort of the reviewers and the comments made to improve the manuscript.  Below I have my response to the Reviewers.  However, if the Editor feels strongly that I should take a different course of action, I will make those changes promptly.

Also, in reviewing the paper once more, I found two places where I provide very minor changes to the text.  I used ‘Track Changes’ so those can be found easily by the Editor.

Reviewer 2 is concerned about the controversy of crediting the pour-through procedure to Yeager et al. (1983), and feels someone with greater authority should decide if that credit is warranted.  Yeager and the group from Virginia Tech are widely credited with this procedure.  There is nothing controversial here.  There is no other person, group, or entity that would claim credit for the procedure in objection to this paper.  This paper only validates the reasons and criteria for properly giving credit.

Reviewer 3 Report

Please find the attached fıle for the comments to the authors.

Author Response

I appreciate the effort of the reviewers and the comments made to improve the manuscript.  Below I have my response to the Reviewers.  However, if the Editor feels strongly that I should take a different course of action, I will make those changes promptly.

Also, in reviewing the paper once more, I found two places where I provide very minor changes to the text.  I used ‘Track Changes’ so those can be found easily by the Editor.

I read through the comments made by Reviewer 3 on the manuscript.  A few suggestions were made to discuss the container volume, however, that discussion took place later in the text.  I believe the Reviewer recognized that fact with a later comment.  I don’t believe any of these changes are necessary, however, I will make them if the Editor believes they should be made.